# Globalization and Economic Stability: An Insight from the Rocket and Feather Hypothesis in Pakistan

Nabila Khurshid [1,*], Chinyere Emmanuel Egbe [2,*], Asma Fiaz [3] and Amna Sheraz [1]

1 Department of Economics, Comsats University, Islamabad 44800, Pakistan
2 Department of Economics and Finance, Medgar Evers College, City University of New York, Brooklyn, NY 11225, USA
3 School of Economics, Quaid-e-Azam University, Islamabad 44800, Pakistan
* Correspondence: nabilakhurshid@comsats.edu.pk (N.K.); egbe@mec.cuny.edu (C.E.E.)

**Abstract:** The purpose of this study was to analyze the irregular pattern of changing inflation as a result of the pass-through of the exchange rate and fluctuations in oil prices in the current globalization scenario. We used annual data sets for crude oil prices, real effective exchange rates, and inflation in Pakistan from 1972 to 2021 for the analysis. The control variables used in the current study were imports (IMP), gross domestic product per capita (GDP), exports (EXP), globalization (GLOB), and interest rates (CRATE). Our findings from a non-linear autoregressive distributed lag (NARDL) analysis showed that inflation had an asymmetric rocket and feather pattern regardless of how globalization was defined or measured. On the other hand, GDP, EXP, and GLOB negatively impacted inflation, and CRATE and IMP had positive effects on inflation. Our study suggested that alternative policies, such as fixing the exchange rate, might decrease uncertainty and stabilize the Pakistani economy in the future. Moreover, increasing the use of sustainable energy would reduce the dependence of the economy on oil prices, which would lower its impact on the economy.

**Keywords:** real effective exchange rate; oil prices; inflation; non-linear autoregressive distributed lag; rocket and feather hypothesis; globalization; Pakistan





## 1. Introduction

Inflation primarily refers to persistent increases in the prices of goods and services. A rise in the prices of commodities leads to a decrease in the purchasing power of the people in society. The decline in the purchasing power of individuals in the economy eventually leads to a reduction in the level of savings and investments in the economy [1]. Oil is an imported good and the exchange rate affects the price of oil immensely. Therefore, understanding the relationship between the exchange rate and the price of oil and its pass-through on inflation is critical for developing economic policies [2]. Fluctuations in exchange rates play a crucial role in shaping the economic link or as a transmission mechanism between domestic and international markets, directly affecting the prices of goods and services in the domestic economy [2]. Exchange rate movements can influence actual inflation and future price movements. Macroeconomists simultaneously linked oil price shocks to diminished economic activity and higher inflation [3]. As Lescaroux and Mignon [4] observed, several transmission networks exist through which oil prices can influence economic activity and inflation. For instance, a surge in the price of crude oil affects the prices of petroleum products, eventually affecting consumers and producers. This is because crude oil prices directly affect the prices of energy-related items, such as household fuels, electricity, and motor fuels. From the production side, a rise in the price of crude oil increases production costs, which lowers the overall output, employment, real wages, investments, and profits.

The government does not fix the exchange rate at a specific level but lets the supply and demand for the currency determine the exchange rate [4]. The participation of the

Central Bank in the foreign exchange market (whether buying or selling dollars) is limited to ensuring orderly conditions and avoiding extreme exchange rate swings [5]. Central banks are also critical to maintaining economic and financial stability. They implement monetary policy to achieve low and stable inflation. Following the global financial crisis, central banks' toolkits for dealing with financial stability risks and managing volatile exchange rates have grown. The policy response of central banks ultimately explains the transmission of the fluctuations in energy prices to general inflation. Theoretically, the degree to which a rise in oil prices gives rise to inflation through complex production costs depends, among other things, on the supply and demand conditions underlying the global oil market and the inflationary expectations of producers and consumers. As inflation expectations rise in the long run, the chances of an increase in energy costs and wages to consumer prices also surge. In other words, rising oil prices may lead to a perpetual increase in general inflation. As oil prices fall, inflationary pressures begin to dissipate [6].

*The Asymmetric Relationship between Oil Prices and Other Variables*

Shocks to the price of oil were blamed for economic recessions, financial crises in different industries, unemployment, depression of investment through uncertainty, high inflation, low equity and bond values, trade deficits, etc. [7]. Furthermore, oil prices can have both real effects and inflationary effects. The impact of oil price shocks on the US economy is asymmetric. Specifically, increases in oil prices tend to have significant adverse effects, while decreases do not produce corresponding positive results [8,9]. Therefore, analyzing the asymmetric effects of oil price shocks in the context in which they affect domestic prices is interesting.

The asymmetric impact of a shock can be broadly subdivided into two types. One is an asymmetry in sign (asymmetry of direction), which means that the effect will be different in direction. The other one is an asymmetry in magnitude (asymmetry of size), which describes a difference in the scale of changes. Several studies [10–13] considered the symmetric impact of oil prices and exchange rate pass-through (EXRPT) on food prices. All these studies addressed the symmetric and linear effects of exchange rate fluctuation. Thus, there exists a research gap in estimating the non-linear impacts of changes in oil prices and their impacts on macroeconomic variables. This is because the macroeconomic variables do not behave linearly in practice if real-time data is used.

Asymmetric relationships between variables suggest that increasing and decreasing variables may have differential effects. For example, rising oil prices may impact inflation differently than falling oil prices. Positive oil price shocks tend to increase the money supply in oil-producing countries [14], significantly affecting consumer prices [15]. Furthermore, falling oil prices reduce the foreign earnings of oil-producing countries, resulting in currency depreciation and rising inflation [16].

Concern regarding asymmetric pricing was explored by researchers during high-priced episodes. Karrenbrock [17] published several studies from the United States during the gasoline price spike caused by Iraq's invasion of Kuwait in 1990. Sen [18] referenced official investigations into anti-competitive practices in Canada and the United States. Bacon [19] noted that the Monopolies and Mergers Commission in the United Kingdom had investigated gasoline pricing for anti-competitive behavior three times and found evidence of asymmetric pricing. The high oil prices that caused the peak in 2008 produced similar responses. A typical reaction was the Attorney General's Report on gasoline pricing in Bocklet and Baek [20] and an investigation into asymmetric pricing in Portugal. Similar responses also occurred in developing countries, as noted by Kojima [21]. Currently, the Russia–Ukraine war has created the same scenario. As stated by a Bloomberg report, thousands of miles away from the Ukraine conflict, Pakistan is facing the strain of the war and the West's push to penalize Russia for its invasion. South Asian countries are experiencing an energy crisis due to a fuel shortage and rising oil prices in the international market. They are enduring prolonged power outages, mainly due to their inability to procure liquefied natural gas (LNG) from Italian and Qatari suppliers.

All of this is the outcome of a European campaign to abandon Russian fuel to isolate Moscow due to its offensive against Ukraine, shifting the entire burden of fuel procurement to suppliers other than Russia. This has caused an oil price hike and led to mayhem in Pakistan. With the increase in oil prices amid the Russia–Ukraine war, the economy of Pakistan could be gravely impacted, causing the country's currency to be devalued, increasing the current account deficit, and spiking inflation [21,22]. With rising oil prices, declining exchange rates, and rising inflation in Pakistan, scholars are researching the subject.

In exchange-rate-related studies, especially in Pakistan, researchers [23–26] empirically investigated the effect of oil price changes and shocks on inflation. In Pakistan, petroleum products account for approximately 50% of the current energy consumption. Moreover, the share of motor spirit (petrol) and high-speed diesel (HSD) exceeds 50% in the petroleum product group. The country's demand for petroleum products is nearly 21 million tons, out of which not even 19% is met through resources that are available locally, while the balance is met through imports. Oil is the most expensive and widely used among the fuels utilized in thermal power plants. Policymakers cannot overlook the fluctuating price of oil and its potential impact on prices because it is inextricably linked to our daily lives in modern times.

Previous studies did not account for the role of exchange rate fluctuations and oil prices to determine the reasons behind asymmetric inflation (rocket and feather hypothesis) in Pakistan. Therefore, the first contribution that this study made was to provide additional evidence on the rocket and feather hypothesis as shown in Pakistan. The rocket and feather hypothesis refers to the asymmetric changes in domestic price indices of exchange rate pass-through, together with fluctuations in the price of oil. Prices do not always change systematically. Hence, the logic behind it is interpreted as the rocket and feather hypothesis [27]. The rocket and feather hypothesis focuses on the rapid change in variables associated with the economy caused by transactions on the real effective exchange rate and fluctuations in oil prices. In other words, the rocket and feather hypothesis analyses the pattern of the consumer price index varying on one side differently to how it varies on the other. A rocket speeding swiftly and a feather drifting slowly are analogous to the asymmetric pattern associated with the movement of the variables because of the oscillations and variations in the economy. The inspection of asymmetric patterns in EXRPT and oil prices was previously absent in Pakistan's case.

This study made three significant contributions to the literature. First, the current research objective was to test whether and how the rocket and feather hypothesis was valid under the forces of globalization. We considered three types of globalization: globalization, economic globalization, and trade globalization. Second, in contrast to the previous studies that assumed that the nature of the variables was linearly cointegrated in the case of Pakistan, we executed suitable non-linear cointegration techniques to address the issue of non-linearity. Specifically, our study used annual data from 1972 to 2021 to examine the asymmetry and non-linearities of exchange rate pass-through and oil price fluctuations on inflation. Finally, our study was the first study to our knowledge that attempted to investigate the impact of the EXRPT and oil price on inflation jointly. Accordingly, we answered the following questions: (1) Is there any evidence of an asymmetrical EXRPT to be found during appreciation and depreciation? (2) Was there any nonsignificant EXRPT in the long run? (3) Was there an asymmetric relationship between oil prices and inflation? To answer these questions, we first estimated the EXRPT and oil price effects on inflation by applying Shin et al.'s non-linear autoregressive distributed lag (NARDL) framework [28].

## 2. Some Glimpses from Previous Literature

Bénassy-Quéré et al. [29] found that the long-run pass-through is significant and inelastic in the short run, having lower pass-throughs than the long-run exchange rate pass-through effect on the consumer price index. Unlike Usman et al. [30], Baharumshah et al. [31] found that the pass-through was more asymmetric in the short run than in the

long run. No changes were observed in appreciation in the short run, while inflation affects consumer price levels, leading to increased depreciation in Sudan. Baryan and Cecchetti; Balcilar et al. [32,33] also found, using the vector smooth transition autoregressive (VSTAR) method, that there was evidence of asymmetric pass-through in all the countries exhibiting a higher pass-through when the size of the shocks to the transition variable moved the system above a threshold level. Delatte and López-Villavicencio [34] discovered that currency depreciations have stronger effects than currency appreciation in the long and short runs. The pass-through effect is higher in countries with large import shares [35]. Jiang and Kim [36] concluded that shocks in the exchange rate have an instant effect, i.e., within one quarter, on producers' prices. The producer price index (PPI) and retail price index (RPI) are generally not complete relative to the exchange rate pass-through (ERPT), PPI and RPI are influenced relatively quickly, and the ERPT's effect on the RPI is less than that of the ERPT on the PPI in China. Younus and Yucel [37] carried out research regarding the pass-through of the exchange rate in Pakistan to inflation from the period of 2008 to 2019. Their result indicated a substantial impact of the exchange rate on domestic price levels, which had an upward trend. Saidu et al. [38] found that a decrease in oil price had a more significant impact on the exchange rate than an increase, and this varied in sign and size across countries. In their research, Kwofie and Ansah [39] concluded that inflation and the stock market have a significant relationship over the long run while showing no relationship in the short run.

On the other hand, the exchange rate has a relationship with the stock market, both in the long and short runs. In the short run, the parallel exchange rate pass-through affects inflation significantly in the long run [40]. If we look at the exchange rate, depreciation of the exchange rate leads to an increase in import prices and an increase in inflation expectations, which leads to a rise in wage demand, thus creating upward inflationary pressure [41].

Milani [42] applied the New Keynesian model using variables, such as IS, and Philips curves to study inflation in the US economy. Their results led them to the conclusion that the sensitivity of US inflation to global measures of output may have increased over the sample, but it remained minimal. Therefore, globalization has a limited effect on the US economy. Sajid and Siddiqui; Fiaz et al. [43,44] studied the effect of money growth and exchange rate depreciation on inflation in Pakistan. They took Pakistani data from 1982 to 2012 and applied the Hansen Model in combination with a Philips curve equation. They evaluated their results by using the generalized moments technique. It was found that money growth affects inflation and the effect on inflation due to exchange rate depreciation is present but is not significant. Algaeed [45] concluded that exchange rate depreciation alongside symmetric price shocks of oil could be responsible for domestic consumer price inflation in the long run in Saudi Arabia. There are threshold effects on the inflation of money growth, but there is no threshold effect on the inflation of exchange rate depreciation [46].

Shahzad and Jaffri [47] and Wimanda [48] investigated whether the exchange rate, output gap, global energy inflation, and lagged inflation positively affect monthly inflation in Pakistan. Based on their results, they suggested that a fixed exchange rate is vital for controlling domestic inflation. Ahmed et al. [49] estimated that the most significant shock in variables is noted in the money supply compared with other variables from the exchange rate, which is the most significant cause of fluctuation in Pakistan's economy. Asad et al. [50] also highlighted a strong and significant positive relationship between inflation and the real effective exchange rate. Ali et al. [51] observed that an increase in interest rates and money supply leads to inflation and exchange rate volatility.

Using quarterly data from 2008 to 2019, Younus and Yucel [37] researched the pass-through of the swapping scale in Pakistan with expansion locally, which strongly affects the customer container, from 2008 to 2019. Ahmed et al. [49] assessed the conversion scale of the pass-through to costs of oil and imported great costs, alongside the purchaser costs, cash supply, and loan fees in Pakistan using monthly data from 2005 to 2015 and calculating it using an unlimited VAR model. These studies observed that the most significant shock in

factors is noted in cash supply when contrasted with different factors from the swapping scale, which is the most crucial reason for the change in the nation's economy.

Benigno and Faia [52] studied the impact of globalization on pass-through channels. Specifically, they used sectoral information for the US to analyze the pass-through of conversion standards, particularly after 1999. The outcomes indicated an increase in the level of pass-through for close to half of the sectors in their study. Their results suggest that in the wake of changing the exchange strategy toward more major advancement, the ERPT rises. Additionally, López-Villavicencio and Mignon [53] analyzed the pass-through of swapping scales to import costs of three central Eurozone nations by focusing on globalization in the pass-through channel. Gust et al. [54] contended that rising unfamiliar rivalry through globalization prompts a lower ERPT. They argued that in an open economy, firms' evaluating choices depend on the host's minor expenses and the contenders' costs. Accordingly, the fall in pass-through is ascribed to bringing down exchange costs because of globalization. A study of Nigeria also provides intriguing indications of the degree of the pass-through of the swapping scale, disregarding that pass-through during the time of enthusiasm for homegrown money is not the same as devaluation.

*Concluding Remarks*

Pakistan is a developing country that experienced very few years of economic stability during the study period. Low growth was caused by various factors, including political unrest, social issues, cultural barriers, and policymakers' poor-quality economic policies. Every country now relies on others for goods and commodities in an era of rapid globalization. Furthermore, the exchange rate and oil prices are critical to economic growth in developing countries, such as Pakistan. By reviewing the past literature, we concluded that the non-linear effect of real effective exchange rate pass-through on prices following the rock and feather hypothesis was missing in the case of Pakistan. Therefore, in this research, the rocket and feather hypothesis was investigated.

## 3. Materials and Methods

### 3.1. Conceptual Framework

Theoretically, the exchange rate affects the economy's exports and imports, which affects the value of the money in the economy and eventually affects the demand and supply of goods. As oil is an imported item, the change in the exchange rate affects the prices of imported goods. International changes in oil prices, along with a change in the exchange rate, have a high impact on oil prices. Oil is used in the production and transportation of goods, which affects the cost of production, ultimately affecting the prices of output, leading to an increase in overall prices in the economy. Inflation affects the economy's purchasing power, affecting the income per capita. Furthermore, globalization affects the demand and supply of goods, which affect the prices in international markets and ultimately affect the prices in the domestic economy. Figure 1 shows the transmission mechanism of the effects of oil prices and EXPRT on food prices, along with other controlled variables.

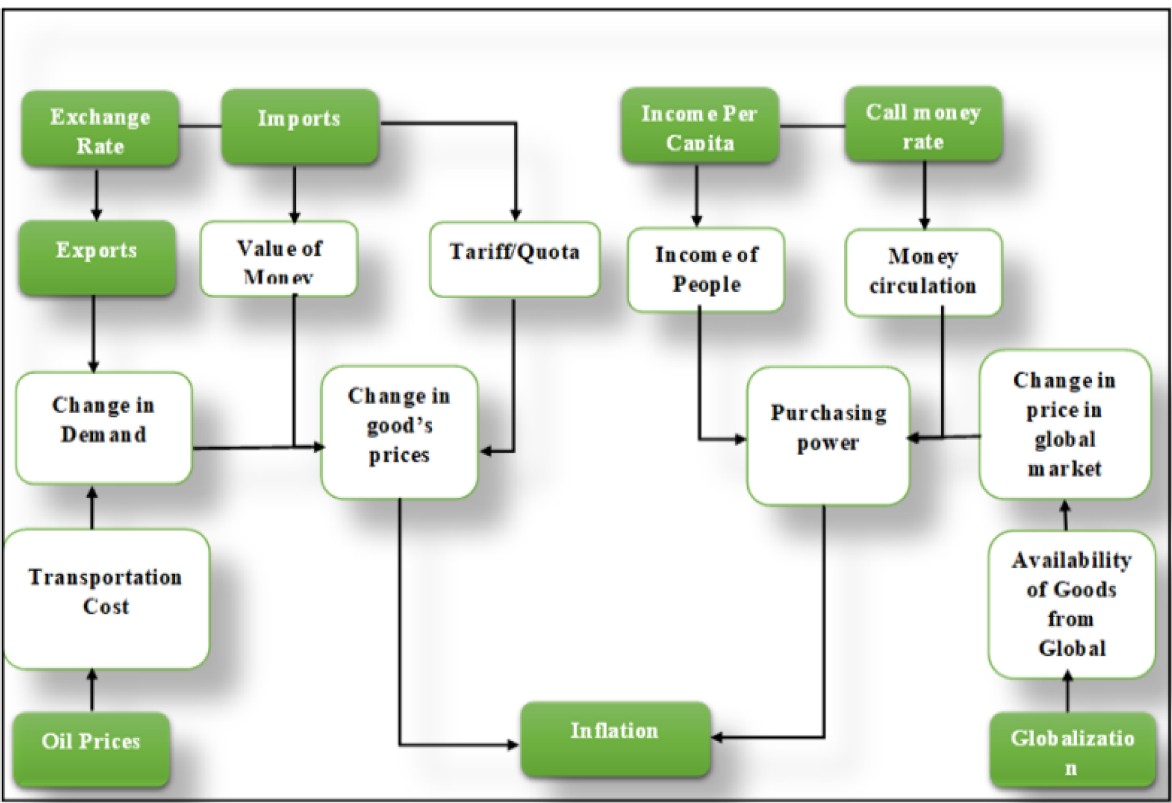

**Figure 1.** EXRPT transmission mechanism from foreign oil prices to local food prices.

*3.2. Econometric Methodology*

3.2.1. Variables, Data Sources, and Econometric Model

For this research study, we used annual data for Pakistan from 1972 to 2009. The consumer price index (CPI) was used as a proxy to measure inflation and is the weighted average of the price of a basket of goods, such as medicine and transportation, that a consumer requires on a day-to-day basis [55]. The details of the variables used in the current research are presented in Table 1.

**Table 1.** Descriptions of the variables.

| Abbreviations | Explanation | Data Source |
|---|---|---|
| Dependent Variable | | |
| INF (inflation) | Consumer price index was used as a proxy for inflation | IFS |
| Independent Variables | | |
| LRER (real effective exchange rate) | The real effective exchange rate is a proxy for the exchange rate | SBP |
| OILP (oil prices) | The spot price of a barrel of crude oil | WDI |
| LIMP (imports) | Imports in billions of US dollars | SBP |
| LEXP (exports) | Export in billions of US dollars | SBP |
| CRATE (interest rate) | The policy rate is a proxy for the interest rate | SBP |
| INCOM (GDP/capita) | GDP per capita | WDI |
| GLOB (globalization) | Globalization index | KOF Globalization Index |
| EGLOB (economic globalization) | Economic globalization index | KOF Globalization Index |
| TGLOB (trade globalization) | Trade globalization index | KOF Globalization Index |

Data for this study was taken from the World Bank Index Stats, the State Bank of Pakistan Statistics, and the Economic Survey of Pakistan. Data on globalization was derived

from the KOF Swiss Economic Index of Pakistan. Using the data we obtained for the study, we estimated the model specified in Equation (1):

$$INF_t = \beta_0 + \beta_1 LRER_t + \beta_2 OILP_t + \beta_3 LIMP_t + \beta_4 EXP_t + \beta_5 CRATE_t + \beta_6 INCOM_t + \beta_7 GLOB_t$$
$$+ \beta_8 EGLOB_t + \beta_9 TGLOB_t + \mu_t$$

(1)

where the subscript *t* refers to a specific time, each $\beta$ is a constant to be determined, and $\mu_t$ is the error term.

3.2.2. Description of the Variables

Dependent Variable

Inflation is measured as the change in the consumer price index (CPI). Inflation refers to the general increase in prices over time within a specific period [55].

Independent Variables

Real Effective Exchange Rate

The exchange rate is the weighted average of the country's currency in a nominal effective manner against other foreign currencies. The effect of the exchange rate is seen on the dependent variable, where its frequencies and fluctuations are observed in inflation [55]. The real value of the exchange rate adjusted for inflation is known as the real effective exchange rate used in most studies:

$$RER = \frac{NER \ x \ price \ of \ domestic \ goods}{price \ of \ foreign \ goods}$$

where *NER* is the nominal exchange rate.

Income Per Capita

Per capita income is the average income earned by individuals living in a specific area or region [31]. It indicates how much an individual earns over a period and helps to measure their purchasing abilities and future trends. It is commonly calculated by using the formula.

$$Income \ per \ capita = \frac{National \ income}{population}$$

Globalization

The globalization index refers to the rate of globalization across the different countries of the world economy by measuring economic integration. The globalization index effects are seen and observed on the consumer price index by observing the world market and trading systems [56].

Economic Globalization

Economic globalization refers to the increasing interdependence of world economies due to the growing scale of the cross-border trade of commodities and services, the flow of international capital, and the wide and rapid spread of technologies. It reflects the continuing expansion and mutual integration of market frontiers and was an irreversible global economic development trend at the turn of the millennium. Economic globalization also refers to the agreement of countries to meet economic goals, which affects components such as the consumer price index.

Trade Globalization

Trade globalization is a measure of economic integration for a country [57]. More generally, it loosely represents the proportion of all production that crosses the boundaries of a country, as well as the number of jobs in that country that are dependent upon external trade. On a global scale, it represents the proportion of all world production used for imports and exports between countries.

Imports

Imports refer to the goods and services a country purchases from other countries. Each country has different import policies, and their imports are also heavily dependent upon the value of their currency.

Exports

Exports are goods and services that a country produces internally and then sells to other countries. A country's exports depend on the exchange rate, which, in turn, depends on the value of its currency in the world market. The exchange rate of a country's currency affects the price of imported goods and the consumer process.

Oil Prices

Oil prices generally denote the spot price of a barrel of crude oil. As crude oil prices fluctuate daily, it affects other economic parameters, such as inflation. Specifically, since the production of manufactured goods depends heavily upon transport costs, increases in the price of oil affect the price of manufactured goods and other production costs.

Call Money Rate

The call money rate is the benchmark of the interest rate established by the banks for brokers who obtain loans for fund purposes. Investors are supposed to pay the call money rate on margin accounts for not facing any losses, which also indirectly affects the CPI [58].

3.2.3. Estimation Technique

The econometric model for our study is specified below in Equations (2) and (3) and estimated using the non-linear autoregressive distributed lag (NARDL) model. The stability of the model was tested using the augmented Dickey–Fuller test. The variables were mixed; some were stationary, while some were nonstationary and integrated using an order of one. According to extant theory, we applied the NARDL methodology in this situation. Following the estimation sequence, we checked the long-run relation view ARDL. The ARDL model was as follows:

$$
\begin{aligned}
INF_t = \eta_0 + \sum_{i=1}^{q} & \eta_1 \left(INF\right)_{t-1} + \sum_{i=1}^{q} \eta_2 \left(RER\right)_{t-i} + \sum_{i=1}^{q} \eta_3 \left(OILP\right)_{t-i} + \sum_{i=1}^{q} \eta_4 \left(LIMP\right)_{t-i} \\
& + \sum_{i=1}^{q} \eta_5 \left(LEXP\right)_{t-i} + \sum_{i=1}^{q} \eta_6 \left(CRATE\right)_{t-i} + \sum_{i=1}^{q} \eta_7 \left(INCOME\right)_{t-i} + \sum_{i=1}^{q} \eta_8 \left(GLOB\right)_{t-i} \\
& + \sum_{i=1}^{q} \eta_9 \left(EGLOB\right)_{t-i} + \sum_{i=1}^{q} \eta_{10} \left(TGLOB\right)_{t-i} + \mu_t
\end{aligned}
\tag{2}
$$

ARDL Cointegration Equation

$$
\begin{aligned}
INF_t = \eta_0 + \sum_{i=1}^{q} & \eta_1 \left(INF\right)_{t-1} + \sum_{i=1}^{q} \eta_2 \left(RER\right)_{t-i} + \sum_{i=1}^{q} \eta_3 \left(OILP\right)_{t-i} + \sum_{i=1}^{q} \eta_4 \left(LIMP\right)_{t-i} \\
& + \sum_{i=1}^{q} \eta_5 \left(LEXP\right)_{t-i} + \sum_{i=1}^{q} \eta_6 \left(CRATE\right)_{t-i} + \sum_{i=1}^{q} \eta_7 \left(INCOME\right)_{t-i} + \sum_{i=1}^{q} \eta_8 \left(GLOB\right)_{t-i} \\
& + \sum_{i=1}^{q} \eta_9 \left(EGLOB\right)_{t-i} + \sum_{i=1}^{q} \eta_{10} \left(TGLOB\right)_{t-i} + \lambda_1(INF)_t + \lambda_2(RER)_t + \lambda_3(OILP)_t \\
& + \lambda_4(LIMP)_t + \lambda_5(LEXP)_t + \lambda_6(CRATE)_t + \lambda_7(INCOME)_t + \lambda_8 \left(GLOB\right)_t \\
& + \lambda_9(EGLOB)_t + \lambda_{10}(TGLOB)_t + \mu_t
\end{aligned}
\tag{3}
$$

Equations (2) and (3) are an error-correction specification that provides both long-run and short-run coefficients. In this model, each λ represents a long-run coefficient, while each difference variable η depicts a short-run coefficient. However, Equations (2) and (3) show the symmetric relationships between explanatory variables. Considering the aspect of non-linearity, which is essential, we are concerned that positive and negative changes in LRER and OILP may have different effects. To capture the asymmetric effect

of these positive and negative changes, the NARDL model is more appropriate (Mundell et al., 2014) [59], where the exchange rate and oil prices were decomposed into LRER_POS, LRER_NEG, OILP_POS, and OILP_NEG. Therefore, the model was as follows:

Real Effective Exchange Rate

$$POS = \sum_{j=1}^{t} lnRER_j^+ = \sum_{j=1}^{t} \max(lnRER, j) \tag{4}$$

$$NEG = \sum_{j=1}^{t} lnRER_j^- = \sum_{j=1}^{t} \min(lnRER, j) \tag{5}$$

Oil Proces

$$POS = \sum_{j=1}^{t} OILP_j^+ = \sum_{j=1}^{t} \max(OILP, j) \tag{6}$$

$$NEG = \sum_{j=1}^{t} OILP_j^- = \sum_{j=1}^{t} \min(OILP, j) \tag{7}$$

By combining these equations, we obtained the following asymmetric error correction equation:

Model 1

$$
\begin{aligned}
INF_t = \alpha_0 + INF_{t-1} \quad &+ \sum \alpha_1 LRER_{t-1}{}^+ + \sum \alpha_1 LRER_{t-1}{}^- + \sum \alpha_2 LIMP_{t-i} + \sum \alpha_3 LEXP_{t-i} \\
&+ \sum \alpha_4 GLOB_{t-i} + \sum \alpha_5 CRATE_{t-i} + \sum \alpha_6 OILP_{t-i}{}^+ + \sum \alpha_6 OILP_{t-i}{}^- + \sum \alpha_7 INCOM_{t-i} \\
&+ \pi_0 + INF_t + \sum \pi_1 LRER_t{}^+ + \sum \pi_1 LRER_t{}^- + \sum \pi_2 LIMP_t + \sum \pi_3 LEXP_t \\
&+ \sum \pi_4 GLOB_t + \sum \pi_5 CRATE_t + \sum \pi_6 OILP_t{}^+ + \sum \pi_6 OILP_t{}^- + \sum \pi_7 INCOM_t + \mu_t
\end{aligned}
$$

Model 2

$$
\begin{aligned}
INF_{t-1} + \sum \alpha_1 LRER_{t-1}{}^+ \quad &+ \sum \alpha_1 LRER_{t-1}{}^- + \sum \alpha_2 LIMP_{t-i} + \sum \alpha_3 LEXP_{t-i} + \sum \alpha_4 EGLOB_{t-i} \\
&+ \sum \alpha_5 CRATE_{t-i} + \sum \alpha_6 OILP_{t-i}{}^+ + \sum \alpha_6 OILP_{t-i}{}^- + \sum \alpha_7 INCOM_{t-i} + \pi_0 + INF_t \\
&+ \sum \pi_1 LRER_t{}^+ + \sum \pi_1 LRER_t{}^- + \sum \pi_2 LIMP_t + \sum \pi_3 LEXP_t + \sum \pi_4 EGLOB_t \\
&+ \sum \pi_5 CRATE_t + \sum \pi_6 OILP_t{}^+ + \sum \pi_6 OILP_t{}^- + \sum \pi_7 INCOM_t + \mu_t
\end{aligned}
$$

Model 3

$$
\begin{aligned}
INF_t = \alpha_0 + INF_{t-1} \quad &+ \sum \alpha_1 LRER_{t-1}{}^+ + \sum \alpha_1 LRER_{t-1}{}^- + \sum \alpha_2 LIMP_{t-i} + \sum \alpha_3 LEXP_{t-i} \\
&+ \sum \alpha_4 TGLOB_{t-i} + \sum \alpha_5 CRATE_{t-i} + \sum \alpha_6 OILP_{t-i}{}^+ + \sum \alpha_6 OILP_{t-i}{}^+ + \sum \alpha_7 INCOM_{t-i} \\
&+ \pi_0 + INF_t + \sum \pi_1 LRER_t{}^+ + \sum \pi_1 LRER_t{}^- + \sum \pi_2 LIMP_t + \sum \pi_3 LEXP_t \\
&+ \sum \pi_4 TGLOB_t + \sum \pi_5 CRATE_t + \sum \pi_6 OILP_t{}^+ + \sum \pi_6 OILP_t{}^- + \sum \pi_7 INCOM_t + \mu_t
\end{aligned}
$$

## 4. Results and Discussion

### 4.1. Stationarity Test

The results of our estimation process are presented in Tables 2–4. The results of the unit root test of stationarity in Table 2 show that the variables used were a mixture of stationary and nonstationary variables. Furthermore, no variable in the model was stationary at the second lag. Therefore, the NARDL non-linear autoregressive lag methodology was appropriate for this estimation. The results of the long-run bound test (Table 2) showed that there existed a non-linear relationship between the variables.

**Table 2.** Unit Root Results.

| Variables | At Level | Integration | Decision |
|---|---|---|---|
| INFLATION | −3.46 ** | −9.46 *** | I(0) |
| INCOM | −0.23(no) | −11.39 *** | I(1) |
| LEXP | −2.94 ** | −6.62 *** | I(0) |
| LIMP | −1.89(no) | −5.68 *** | I(1) |
| LRER | −1.03(no) | −5.62 *** | I(1) |
| LTOT | −0.76(no) | −8.42 *** | I(1) |
| OILP | −1.61(no) | −6.36 *** | I(1) |
| PORATE | −2.14(no) | −5.42 *** | I(1) |
| GLOB | −1.13(no) | −4.73 *** | I(1) |
| CRATE | −3.08 ** | −6.58 *** | I(0) |
| EGLOB | −2.04(no) | −6.52 *** | I(1) |
| TGLOBA | −1.84(no) | −7.50 *** | I(1) |

Note: (***) Significant at the 1%; (**) Significant at the 5% and "no" means Not Significant.

### 4.2. Bound Test

Table 3 shows the results of the bound test of the three models developed based on globalization, economic globalization, and trade globalization. The bound test shows the upper and the lower limit at each level of significance; it confirmed the long-run relationships between the variables, as the F-statistic value was greater than the limit that validated our variables and the study. It showed the significance of the variables used in the study and their impact on inflation.

**Table 3.** Bound Test Results.

| Bound Test Value | Signif. | I(0) | I(1) |
|---|---|---|---|
| 11.781 | 10% | 3.02 | 2.05 |
| 7.614 | 5% | 3.33 | 2.3 |
| 10.904 | 2.50% | 3.6 | 2.52 |
| | 1% | 3.93 | 2.79 |

### 4.3. Empirical Results

It is well understood in international economic transactions that finance, trade, tourism, investment, migration, and many more are all influenced by cross-border monetary policies. The governments of many third-world countries have actively searched for solutions that can help them to overcome the instability in international currency markets. Their policymakers have explored or experimented with different currency schemes, such as dollarization, managed floating exchange rates, nominal anchors, and target bands. Hypothetically, in an open economy, capital flexibility profoundly affects the policy choices of the exchange rate. As Robert Mundell explained, the management of a monetarily unified economy faces a choice between monetary policy autonomy or a fixed exchange rate [57]. Empirically, the impact of globalization has been observed in the international exchange rates for quite a few years. With the increasing trend of globalization, advanced trading countries are less prone to adverse shocks from unstable oil prices compared with less developed countries and newly industrialized countries (NICs). An increase in the price of oil results in a comparatively lower rate of economic loss over time for the advanced trading nations. The differences in the impact of fluctuating or rising oil processes may be because the advanced trading nations are gradually becoming effectively information-based and less energy-intensive in the presence of globalization. On the other hand, with the development of heavy industrialization, NICs are in their initial phase of becoming more energy intensive. Therefore, they might be expected to be more subject to the adverse impacts that are associated with the increased energy price.

### 4.3.1. Models of Globalization and the Rocket and Feather Hypothesis

In this section, the results of three tested models are presented and then summarized in Table 4. Specifically, the non-linear ARDL model was applied to test the rocket and feather hypothesis. All the indicators tested showed significant impacts on inflation.

In model 1, we tested the rocket and feather hypothesis in the presence of globalization. The results of model 1 showed that when the real effective exchange rate decreased, the consumer price index showed a decreasing trend of up to 14.31 percent. In contrast, if the exchange rate increased, the consumer price index decreased by 10.56 percent. These results demonstrated an asymmetric effect, though the CPI decreased under both conditions (increasing real exchange rate and decreasing real exchange rate). Thus, the rocket and feather hypothesis did make sense in the presence of globalization in the economic system of Pakistan. Due to globalization, fluctuations in oil prices also affected the consumer price index in Pakistan. Specifically, as the price of oil increased, the consumer price index also increased by 0.18 percent, and when the oil prices decreased, the consumer price index increased by 0.02 percent. This result lent support to the asymmetric effect on the consumer price index of the rocket and feather hypothesis. In our literature survey, we observed that LRER_NEG and OILP_NEG inflation is in control as consumer prices are lower in such cases [58]. As a result, the stabilization of the exchange rate and oil prices must be emphasized. When the currency appreciates, inflation tends to decrease more than when the currency depreciates. Therefore, appropriate policies should be developed to cope with the increasing inflation [59].

Model 2 considered the presence of economic globalization. The results indicated that when the real effective exchange rate decreased, inflation decreased by 54.97 percent, whereas if the real effective exchange rate increased, inflation decreased by 2.92 percent. Therefore, it can be noted that in both scenarios of exchange rate fluctuations, inflation fell. However, as the exchange rate increased, inflation moved toward being positive. These results provided evidence of asymmetric exchange rate pass-through on inflation and lent further support for the rocket and feather hypothesis, as prices decreased more rapidly than they increased. As far as oil prices were concerned, the fluctuation was monitored, along with the exchange rate on the consumer price index to witness the effects. Therefore, the literature suggests that currency appreciation decreases inflation more than it does when the currency depreciates, which also suggests incorporating policies that increase the foreign attraction to local currency [60]. Oil prices tend to decrease inflation in both cases, though the decrease in oil prices supports a reduction in inflation [61].

The rocket and feather hypothesis was further tested using the NARDL in the context of trade globalization. In model 3, we tested how the real effective exchange rate pass-through and fluctuations in the oil prices affected inflation. The results are presented in Table 4. The results showed that when the real effective exchange rate decreased, inflation decreased by 42.72 percent, whereas when the real effective exchange rate increased, the consumer price index also decreased by 62.95 percent. Therefore, in either case, a negative pattern of inflation was observed.

Further results indicated that when the price of oil increased, inflation increased by 0.158 percent. On the other hand, if the price of oil decreased, inflation increased by 0.047 percent. Control variables, such as INCOM, LEXP, and globalization, affected inflation negatively, while interest rates positively impacted inflation in all three models. These results suggested that the stabilization of the exchange rate is crucial in reducing uncertainty and controlling or managing inflation levels in the country [62].

It can be observed from our tests that no matter what the exchange rate trend was, inflation tended to decrease, though at a different rate. Therefore, fluctuations in oil prices can create uncertainties in economic operations. Accordingly, renewable energy resources are being emphasized by countries to decrease their burden in world affairs [63].

**Table 4.** Long-run results.

| Variables | Model 1 | Model 2 | Model 3 |
|-----------|---------|---------|---------|
| | Coefficient | Coefficient | Coefficient |
| LRER_POS | −10.56 | −2.92 | −62.95 |
| | (0.00) | (−0.88) | (−0.04) |
| LRER_NEG | −14.31 | −54.97 | −42.72 |
| | (−0.06) | (0.00) | (0.00) |
| OILP_POS | 0.18 | 0.11 | 0.15 |
| | (0.00) | (0.00) | (−0.01) |
| OILP_NEG | 0.02 | 0.20 | 0.05 |
| | (−0.48) | (−0.01) | (−0.09) |
| INCOM | −8.07 | −19.10 | −8.27 |
| | (−0.10) | (0.00) | (0.00) |
| CRATE | 0.92 | 1.76 | 0.98 |
| | (0.00) | (0.00) | (−0.06) |
| LEXP | −15.22 | −13.96 | −15.33 |
| | (0.00) | (−0.01) | (0.01) |
| LIMP | 11.01 | 2.74 | 11.62 |
| | (−0.03) | (−0.49) | (−0.01) |
| GLOB | −0.81 | - | - |
| | (−0.05) | - | - |
| EGLOB | - | −0.70 | - |
| | - | (−0.09) | - |
| TGLOB | - | - | −0.29 |
| | - | - | (−0.38) |

Note: values in parentheses show the $\rho$ values.

### 4.3.2. Short-Run Results

Table 5 shows the short-run results. The model 1 results show that the variables exports, imports, real effective exchange rate, and globalization were statistically significant, indicating that they impacted inflation. At the same time, the coefficient of oil prices was insignificant. Apart from this, income per capita, exports, and globalization negatively impacted inflation. However, imports and call money rates positively impacted inflation by increasing its magnitude.

Moreover, due to the increase in globalization, the real effective exchange rate fell further. It generated a negative value, while when globalization decreased, the real effective exchange rate increased toward a less negative value. On the other hand, oil prices increased when globalization increased and decreased when globalization decreased.

The results of model 2 showed that the income per capita, exports, and call money rates were statistically significant, whilst imports were insignificant. Therefore, the effects of the exchange rate pass-through were witnessed in the consumer price index. Other variables, such as income per capita, exports, and globalization, had a negative relationship with inflation, while imports and call money rates positively impacted inflation by increasing its magnitude. According to model 3, the statistically significant variables included income per capita, imports, exports, and call money rates, alongside exchange rate and oil prices; thus, they affected inflation accordingly. Moreover, the variables followed a similar trend in trade globalization as in economic and just globalization. Additionally, the results for ECM were also calculated and shown in the table, where the models were highly significant.

**Table 5.** Short-run results.

| Variables | Model 1 Coefficient | Model 2 Coefficient | Model 3 Coefficient |
|---|---|---|---|
| LRER POS (−1) | −131.64 | −4.99 | −83.96 |
| | (0.00) | (0.88) | (0.04) |
| LRER NEG (−1) | −17.85 | −93.81 | −56.99 |
| | (0.06) | (0.00) | (0.00) |
| OILP POS (−1) | 0.22 | 0.18 | 0.21 |
| | (0.00) | (0.00) | (0.01) |
| OILP NEG (−1) | 0.03 | 0.33 | 0.06 |
| | (0.49) | (0.01) | (0.10) |
| LRER_NEG | −0.28 | −16.04 | −7.79 |
| | (0.98) | (0.21) | (0.51) |
| LRER_NEG (−1) | 26.16 | 55.72 | 48.58 |
| | (0.02) | (0.01) | (0.00) |
| LRER_POS | 9.21 | 58.13 | 32.52 |
| | (0.66) | (0.03) | (0.21) |
| LRER_POS (−1) | 104.07 | 82.08 | 87.08 |
| | (0.00) | (0.00) | (0.00) |
| LRER_POS (−2) | 84.26 | 73.24 | 60.14 |
| | (0.00) | (0.01) | (0.00) |
| OILP_POS (−1) | 0.01 | −0.14 | −0.08 |
| | (0.84) | (0.05) | (0.24) |
| OILP_POS (−1) | −0.08 | 82.08 | −0.12 |
| | (0.12) | (0.00) | (0.09) |
| INCOM | −10.06 | −32.6 | −11.03 |
| | (0.11) | (0.01) | (0.00) |
| LEXP | −18.98 | −23.82 | −20.45 |
| | (0.00) | (0.01) | (0.01) |
| LIMP | 13.73 | 4.67 | 15.5 |
| | (0.02) | (0.48) | (0.01) |
| CRATE | 1.15 | 3.00 | 1.31 |
| | (0.00) | (0.00) | (0.07) |
| GLOB | −1.01 | - | - |
| EGLOB | - | −1.19 | - |
| | | (0.11) | |
| TGLOB | | - | −0.39 |
| | | | (0.38) |
| ECM | −0.92 | −0.87 | −0.92 |
| | (0.00) | (0.00) | (0.00) |

Note: values in parentheses in the model columns show the ρ values.

### 4.4. Granger Causality Test

Table 6 shows the results of the Granger causality test. It was observed that unidirectional causal relationships were found for LRER, OILP, INCOM, LEXP, LIMP, CRATE, GLOB, EGLOB, and TGLOB. Hence, these causal relationships supported the elasticity of the NARDL for each series.

**Table 6.** Granger causality results.

| Null Hypothesis | F-Statistic | Prob. |
|---|---|---|
| INCOM does not Granger cause INFLATION | 3.16018 | 0.0529 |
| INCOM does not Granger cause LIMP | 11.4462 | 0.0001 |
| LRER_NEG does not Granger cause INCOM | 2.53454 | 0.0914 |
| OILP does not Granger cause INCOM | 2.44136 | 0.099 |
| INCOM does not Granger cause EGLOB | 7.39790 | 0.0017 |
| LRER_POS does not Granger cause LEXP | 3.45912 | 0.0407 |
| LIMP does not Granger cause LRER_POS | 2.51930 | 0.0926 |
| LRER_NEG does not Granger cause LIMP | 2.51974 | 0.0926 |
| CRATE does not Granger cause LIMP | 5.10313 | 0.0103 |
| LIMP does not Granger cause CRATE | 4.87031 | 0.0124 |
| LRER_POS does not Granger cause LRER_NEG | 3.07807 | 0.0566 |
| OILP does not Granger cause LRER_POS | 4.64383 | 0.0151 |
| OILP does not Granger cause CRATE | 3.08965 | 0.0558 |
| CRATE does not Granger cause EGLOB | 4.14228 | 0.0226 |

*4.5. Structural Stability Test*

Figures 2–4 show the graphs of the models that represented the CUSUM and the CUSUM square of the plot to check their stability for the coefficients, both in the long run and the short run of two different models. These figures describe the presence of globalization in the economy, which was found to be under the critical limit of 0.05 and predicted structural stability and goodness of fit.

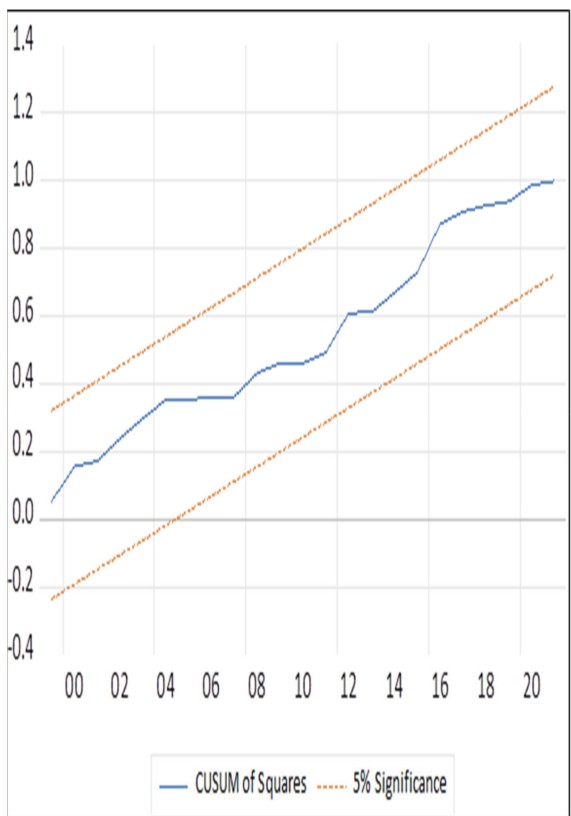

**Figure 2.** CUSUM globalization.

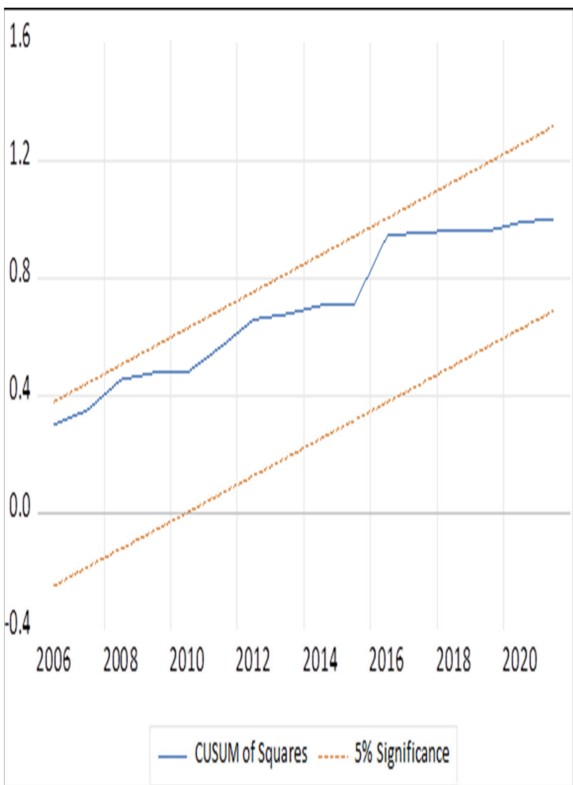

**Figure 3.** CUSUM economic globalization.

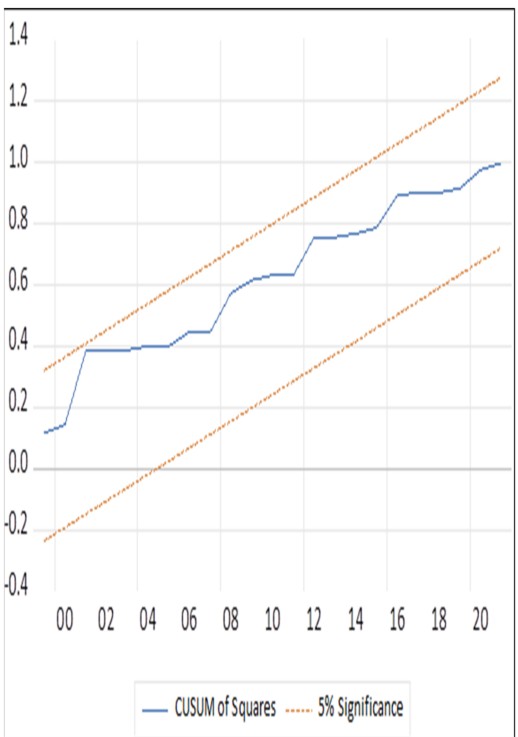

**Figure 4.** CUSUM trade globalization.

Table 7 provides information about the diagnostic testing of the data. The table indicates that there was no serial correlation between the variables because the probability value was greater than 0.05, which led us to reject the null hypothesis that there was a serial correlation in the data. The second diagnostic test was White's test for the detection

of heteroskedasticity. These results also showed no signs of heteroskedasticity among the variables, as the probability values found using White's test were greater than 0.05, which meant that we rejected the null hypothesis.

**Table 7.** Diagnostic tests.

| Items | Test Applied | Value | Prob. |
| --- | --- | --- | --- |
| Serial correlation | Breusch–Godfrey LM test (F-stat) | 3.089 | 0.154 |
| Heteroskedasticity | White's test (F-stat) | 1.598 | 0.517 |
| Normality | Histogram test (Jarque–Bera) | 0.942 | 0.624 |

## 5. Conclusions and Policy Implications

Pakistan is a developing country that has witnessed very few years of economic stability during the period covered by this study. The low growth was due to many reasons, including political unrest, social issues, cultural barriers, and poor-quality economic policies by policymakers. In a time of rapid globalization, every country now depends on others for goods and commodities. If one country changes its policies, it quickly affects the economy of another. Moreover, in the case of developing countries, such as Pakistan, the main component of growth, namely, oil, is mainly imported from other countries. Therefore, the economic systems of developing nations suffer from having a lower ability to contribute or affect prices. The real effective exchange rate pass-through and the fluctuations in oil prices show significantly different effects on inflation that can be observed. Therefore, in this study, the evidence of the rocket and feather hypothesis was explored in the scenario of globalization in the case of Pakistan. It was observed that the asymmetric exchange rate pass-through led to the domestic economy inflation increasing similar to a rocket when the oil prices increased and falling similar to feathers when oil prices decreased. The data set used for empirical analysis was from 1972 to 2021 and applied the non-linear autoregressive distributed lag (NARDL) framework. The variables included annual crude oil prices, real effective exchange rates, imports, gross domestic product per capita, exports, globalization, interest rate, and inflation. The results showed that INCOME, EXP, and GLOB negatively impacted inflation, as an increase in income with an increase in exports and globalization led to more production, which eventually reduced inflation. However, CRATE and IMP produced a positive impact on inflation. An increase in the interest rate led to reduced investment, which reduced production, and when production was lower, the demand increased prices in the economy.

Furthermore, the increase in imports negatively impacted domestic production and demand for domestic products, which negatively affected inflation. As far as oil prices were concerned, the fluctuation was also monitored, along with the exchange rate on the consumer price index, to witness the effects. The results suggested that currency appreciation decreased inflation more than it did when the currency depreciated, which further suggested incorporating policies that increase foreign attraction to the local currency [64,65]. Oil prices tend to decrease inflation in both scenarios, though the decrease in oil prices supports a reduction of inflation [66]. This result lent support to the asymmetric effect on the consumer price index of the rocket and feather hypothesis. The results showed that LRER_NEG and OILP_NEG inflation was in control as the consumer prices were decreased in such cases [67]. Therefore, the results indicated that stabilizing the exchange rate and oil prices must be emphasized for economic growth. When the currency appreciates, inflation tends to decrease more than when the currency depreciates. Thus, appropriate policies should be developed to cope with increasing inflation [68]. The rocket and feather hypothesis model of the real effective exchange rate and the oil price fluctuations on the inflation were verified by the results. Thus, the country's policies should make a subtle shift and work on maximizing the purchasing power of individuals to stabilize the economic system as much as possible. Policies such as fixing the exchange rate might provide the much-required rest from the fluxes and stabilize the inflation in the economy, as the increasing pattern of the exchange rate indicated rising inflation, as well as this study's results. This

will further help to stabilize the balance of payment by giving a boost to exports. Additionally, Pakistan's economy also requires support for the practice of sustainable energy in the economy that will increase the use of renewable energy in production and other operations. Lower utilization of oil will decrease the price fluctuations caused by inflation, as the oil price fluctuation impacts inflation significantly, as shown in the results; thus, reducing its usage will have favorable results.

**Author Contributions:** Conceptualization, N.K.; Methodology, N.K. and C.E.E.; Software, N.K. and A.F.; Formal analysis, N.K. and A.F.; Resources N.K. and A.S.; Data curation, N.K. and A.F.; Writing—original draft preparation, N.K. and A.S.; Writing—review and editing, C.E.E., N.K. and A.F; Visualization, N.K. and A.S.; Supervision, N.K; Project administration, N.K. and C.E.E. All authors have read and agreed to the published version of the manuscript.

**Funding:** This research received no external funding.

**Institutional Review Board Statement:** Not applicable as our study was conducted on secondary time series data.

**Informed Consent Statement:** Not applicable.

**Data Availability Statement:** Data will be available on request.

**Conflicts of Interest:** The authors declare no conflict of interest.

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
