# Peer review of "Globalization and Economic Stability: An Insight from the Rocket and Feather Hypothesis in Pakistan"

_sustainability, doi:10.3390/su15021611_

Round 1

Reviewer 1 Report

1) The format is very non-standard, for example, some table titles are placed in the center, some table titles are placed in the right alignment, please standardize the format of the article.

2) Abbreviations and acronyms should be defined for the first time when they are used in the article. The definition of the acronym “CRATE” is not given in the abstract, and many abbreviations in the text are not defined.

3) In section3.1, you should simply explain the picture of the Conceptual Framework.

4) In section 4, you should number each subheading like 4.1, 4.2, etc

5) The whole inferential process is not rigorous enough, and the conclusion is too short to express clearly enough.

Author Response

Dear Reviewer

Thank you so much for your valuable comments. Really your comments help a lot to improve manuscript. here is response of your valuable comments:

The format is very non-standard, for example, some table titles are placed in the center, some table titles are placed in the right alignment, please standardize the format of the article

Needful has been done throughout the paper.

Abbreviations and acronyms should be defined for the first time when they are used in the article. The definition of the acronym “CRATE” is not given in the abstract, and many abbreviations in the text are not defined

Abbreviations have been checked thoroughly.

In section3.1, you should simply explain the picture of the Conceptual Framework

Added now.

In section 4, you should number each subheading like 4.1, 4.2, etc

Required changes have been done

The whole inferential process is not rigorous enough, and the conclusion is too short to express clearly enough

Required changes have been done accordingly.

Reviewer 2 Report

I think this article is rather interesting.

I have just several notes most of them are about some formalities which can be easily fixed.

19 and 22

You use the abbreviation INCOM at 19 and the abbreviation INCOME at 22. Obviously, you are talking about the same thing so you should use the same abbreviation. Also, to my mind this is not a good abbreviation choice. The term “Income” is not equal to the term “gross domestic product per capita”. You can use the abbreviation GDP or GDPpC.

23

You forgot to explain what the abbreviation CRATE means. You should add it at 20 after the words Interest rates.

In the abstract you should write more straightforward the main elements: the object of your study, the main objective and research questions or hypotheses, methods and results.

Introduction

As you use the Rocket and Feather theory as a part of the title, I suggest to add several sentences about this theory and its implementation in other researches.

Also you should use the same style of naming the theory: either “Rocket and Feather”  with capital letters or “rocket and feather” without them.

79

What is EXPT?

139

What is EXRPT?

150-155

What is ERPT?

188

Why “U.S.” is written with dots?

232

I guess some conclusion on your Literature Review is needed here. What problems were not reviewed? Why your new research and your article is new and important?

234

First of all, that’s not OK to have a whole paragraph without any text and only with a scheme.

Second of all, this Creation needs some explanations. Usually the line with an arrow means the process or influence. Your picture causes a lot of questions. Why do you use an arrow connecting Exchange Rate and Exports?  Does Exchange Rate somehow influence the Exports? And why you connect Exchange Rate and Imports with a simple line? How Imports influence the Value of Money? And much more questions…

You declare this picture as your as your Conceptual Framework, so it should be obvious and clear for your reader. For now – it is not.

502-503

“This is due to many reasons that include, including political unrest, social issues, cultural barriers, etc”.

You should rewrite this “… include, including…” part.

Author Response

Dear Reviewer

Thank you so much for your valuable comments. Attached below the response of your valuable comments:

Comments

Response

1

You use the abbreviation INCOM at 19 and the abbreviation INCOME at 22. Obviously, you are talking about the same thing so you should use the same abbreviation. Also, to my mind this is not a good abbreviation choice. The term “Income” is not equal to the term “gross domestic product per capita”. You can use the abbreviation GDP or GDPpC

Needful has been done.

2

You forgot to explain what the abbreviation CRATE means. You should add it at 20 after the words Interest rates.

Added now

3

In the abstract you should write more straightforward the main elements: the object of your study, the main objective and research questions or hypotheses, methods and results.

Change now

4

As you use the Rocket and Feather theory as a part of the title, I suggest adding several sentences about this theory and its implementation in other research.

Also you should use the same style of naming the theory: either “Rocket and Feather”  with capital letters or “rocket and feather” without them.

Rocket and Feather is replaced with rocket and feather throughout the paper.

5

232

I guess some conclusion on your Literature Review is needed here. What problems were not reviewed? Why your new research and your article is new and important?

Concluding remarks were added in literature review.

234

First of all, that’s not OK to have a whole paragraph without any text and only with a scheme.

Second of all, this Creation needs some explanations. Usually the line with an arrow means the process or influence. Your picture causes a lot of questions. Why do you use an arrow connecting Exchange Rate and Exports?  Does Exchange Rate somehow influence the Exports? And why you connect Exchange Rate and Imports with a simple line? How Imports influence the Value of Money? And much more questions…

You declare this picture as your as your Conceptual Framework, so it should be obvious and clear for your reader. For now – it is not.

Now discussion is added along with conceptual framework.

502-503

“This is due to many reasons that include, including political unrest, social issues, cultural barriers, etc”.

You should rewrite this “… include, including…” part.

Changes have done accordingly.

Reviewer 3 Report

- Line 6: Professor

- Line 27: Please avoid abbreviations in keywords; inflation needs to be written in lower case

- Line 70: ...effects according...

- Please avoid words like "many" (line 91) - please be specific

- Please always indicate the year of publications - e.g. line 155: Shin et al. year...

- Line 210/211: please unbold

- Line 233: ?

- Line 250: Why are all these dots there?

- Line 489: That's one figure

- Line 500: Conclusions

- References: please check the referencing style

Author Response

Dear Reviewer

Thank you so much for your valuable comments which help a lot to improve manuscript. below is the response of your valuable comments:

1

Line 27: Please avoid abbreviations in keywords; inflation needs to be written in lower case

Change accordingly.

2

 Line 70: ...effects according...

Sentence has been corrected accordingly.

3

- Please avoid words like "many" (line 91) - please be specific

Done accordingly

4

Please always indicate the year of publications - e.g. line 155: Shin et al. year...

We incorporated this reference as per journal requirement.

5

Line 210/211: please unbold.

Done accordingly

6

Line 233: ?

Deleted the dubling in line 233.

7

 Line 250: Why are all these dots there?

Removed all the dots.

8

Line 489: That's one figure

Yes, now it is merged in one figure.

9

 Line 500: Conclusions

Added accordingly

10

References: please check the referencing style

Reference style is now according to the journal.

Round 2

Reviewer 1 Report

Accept

Author Response

Dear Editor

We have incorporated all your valuable comments. But for English language editing we have submitted paper for editing but due to Christmas Eve, we have to wait for few days. once editing will be done we will submit the paper to journal.

Once again thank you so much for your valuable comments which improve our paper a lot.

Regards

Nabila Khurshid, PhD